# What Cause Large Spatiotemporal Differences in Carbon Intensity of Energy-Intensive Industries in China? Evidence from Provincial Data during 2000–2019

**DOI:** 10.3390/ijerph191610235

**Published:** 2022-08-17

**Authors:** Xin Xu, Yuming Shen, Hanchu Liu

**Affiliations:** 1College of Resources, Environment & Tourism, Capital Normal University, Beijing 100048, China; 2Institutes of Science and Development, Chinese Academy of Sciences, Beijing 100190, China

**Keywords:** energy-intensive industries, carbon intensity, spatiotemporal differences, driver, spatial spillover, spatial econometric model

## Abstract

China has been reported as the world’s largest carbon emitter, facing a tough challenge to meet its carbon peaking goal by 2030. Reducing the carbon intensity of energy-intensive industries (EIICI) is a significant starting point for China to achieve its emission reduction targets. To decompose the overall target into regions, understanding the spatiotemporal differences and drivers of carbon intensity is a solid basis for the scientific formulation of differentiated regional emission reduction policies. In this study, the spatiotemporal differences of EIICI are described using the panel data of 30 provinces in China from 2000 to 2019, and a spatial econometric model is further adopted to analyze its drivers. As indicated by the results: (1) from 2000 to 2019, China’s EIICI tended to be reduced continuously, and the spatial differences at the provincial and regional levels expanded continuously, thus revealing the coexistence of “high in the west and low in the east” and “high in the north and low in the south” spatial patterns. (2) There is a significant spatial autocorrelation in the EIICI, characterized by high and high agglomeration and low and low agglomeration types. Moreover, the spatial spillover effects are denoted by a 1% change in the local EIICI, and the adjacent areas will change by 0.484% in the same direction. (3) Technological innovation, energy structure, and industrial agglomeration have direct and indirect effects, thus affecting the local EIICI and the adjacent areas through spatial spillover effects. Economic levels and firm sizes only negatively affect the local EIICI. Environmental regulation merely has a positive effect on adjacent areas. However, the effect of urbanization level on EIICI has not been verified, and the effect of urbanization level on the EIICI has not been verified. The results presented in this study show a scientific insight into the reduction of EIICI in China. Furthermore, policymakers should formulate differentiated abatement policies based on dominant drivers, spatial effects, and regional differences, instead of implementing similar policies in all provinces.

## 1. Introduction

Climate change has been found as a major challenge facing human beings currently, and coping with climate change turns out to be a global consensus [1,2,3]. China has been the largest carbon emitter and plays a vital role in global climate governance. In response to the Paris Conference on Climate Change in 2015, at important meetings (e.g., the General Debate of the United Nations General Assembly and the Climate Ambition Summit in 2020), the Chinese government has proposed for the first time to achieve a carbon peak by 2030, and carbon neutrality by 2060. To be specific, carbon intensity, as a vital indicator, needs to fall to 65% of the 2005 base year by 2030. Carbon intensity reveals the resource utilization efficiency and carbon emission efficiency in economic development while reflecting the production technology efficiency level of a country or region to a certain extent [4,5]. Emission-reduction and cleaner production policies based on carbon intensity facilitate the formation of a backward-forced and long-term mechanism to boost China’s economic transformation [6]. In 2019, China’s carbon intensity was 48% lower than in 2005, and 17% below the 2030 target [7]. Since China has vigorously adjusted its industrial structure over the past decade, which has led to significantly reduced carbon intensity, there has been limited room for reducing carbon intensity through industrial restructuring by 2030 [8]. Accordingly, China is facing greater challenges to achieve its carbon intensity target in 2030.

The emission reduction effects of different industries play an important role in achieving the carbon peaking and carbon neutrality goals. The carbon intensity of energy-intensive industries (EIICI) is significantly higher than that of other industrial sectors [9,10], which is recognized as the top priority of China’s future emission reduction actions. In accordance with the “National Economic and Social Development Statistical Bulletin of China in 2019”, energy-intensive industries include six sub-sectors including non-metallic minerals, petroleum coking, electric power, chemicals, steel, and non-ferrous metals. Since 2000, China has entered a stage of rapid urbanization development. Activities (e.g., urban development, infrastructure construction, and the transformation of residents’ lifestyles) all require the support of considerable basic products (e.g., cement, steel, petrochemicals, as well as electricity) [11]. The extent of CO_2_ emissions from energy-intensive industries continues to increase. From 2000 to 2015, six energy-intensive industries increased their carbon emissions at an average annual rate of 8.54%. However, energy-intensive industries have much higher emissions than other industries due to their huge demand for primary or secondary energy to produce products. China’s six energy-intensive industries account for more than 50% of the country’s energy-related CO_2_ emissions [12]. The proportion of the total industrial output value of energy-intensive industries in the entire industrial sector has remained at nearly 30% in the long term, whereas the CO_2_ emissions accounted for approximately 80% of the entire industrial sector [9]. The EIICI in 2019 was 2.5 times that of the entire industrial sector. From the perspective of China’s development in the 10 years ahead, it will take a relatively long time for China to complete industrialization and urbanization, which requires energy-intensive industries to continue to provide energy and raw materials. From a foreign perspective, with the advancement of “The Belt and Road” construction [13], whether it is the supporting ship construction and port construction required by the “Maritime Silk Road”, or the railway plans, airport projects, and highways driven by the “Land Silk Road”, the growing demand in the above international markets will significantly stimulate the expansion of China’s energy-intensive industries. Judging by the development trend at home and abroad, China’s energy-intensive industries will occupy an even larger share in the long term. Meanwhile, if EIICI is maintained at a high level for a long time, the greenhouse gases it produces will directly endanger people’s health and quality of life [10,14]. Thus, to achieve the goal of reducing carbon intensity in 2030, China should formulate effective policy measures in energy-intensive industries [15].

It is a systematic project for China to achieve its carbon intensity target by 2030, and this arduous national task must be broken down into provinces. However, China has a vast territory, and there are large regional differences in natural resource endowments, historical foundations, and technical conditions [16]. As a result, there are significant spatial differences in carbon intensity and its influencing mechanism and this imbalance process is increasing [17]. Accordingly, the conclusions obtained only from the research at the overall level of the country are difficult to apply to the development of different regions, which has become a significant issue that has attracted wide attention from the government and scholars. Therefore, focusing on reducing the EIICI, using the panel data of 30 provinces in China from 2000 to 2019, this paper characterizes the temporal and spatial differences of EIICI. On this basis, further considering the spatial spillover effect, several spatial econometric models are established to reveal the driving factors causing the differences. Hopefully, the research results can provide a reference for government departments to formulate and implement differentiated and targeted regional policies.

## 2. Literature Review

Understanding the spatiotemporal differentiation of carbon emission (intensity) is a solid basis for the scientific formulation of differentiated regional emission reduction policies. Extensive studies have been conducted on drivers (e.g., economic level, technological progress, urbanization, and industrial structure) using various methods (e.g., IDA, SDA, and STIRPAT) from different spatial scales (e.g., global, national, and regional scales), and these studies have achieved fruitful results.

There have been differences in carbon emission (intensity) at different spatial scales, which is a common phenomenon and has still been a hotspot of international concern over the past few years. At the global level, the research objects primarily consist of multiple nations [18,19], OECD nations [20], BRICS nations [21], developing nations and developed nations [22], etc. The results show that due to the differences in the development stage, economic development level, energy structure, industrial structure, and other factors of various nations, there is a huge gap in carbon emission (intensity), nations with high-income levels and urbanization rates, and a high proportion of service industries tend to have lower carbon intensity. Since China has been the largest carbon emitter worldwide, its carbon emission changes and emission reduction effects have aroused wide attention. In recent years, there has been an increasing number of studies on China’s carbon intensity from the perspective of spatial differences and spatial effects. Regardless of provincial or regional differences, China’s carbon intensity exhibits significant spatial differentiation characteristics [23,24], which are revealed by the characteristics of “high in the west and low in the east”, “high in underdeveloped areas, as well as low in developed areas” [25]. In addition, from the perspective of spatial effects, the spatial agglomeration characteristics of China’s carbon intensity are significant, and a positive spatial autocorrelation characteristic is identified [26]. From the perspective of different industries, some scholars have studied the spatiotemporal pattern of the carbon intensity of various industrial sectors in China from the manufacturing industry, power industry, petrochemical industry, and cement industry [27,28,29]. As revealed by the results, there are different degrees of spatial differentiation in various industrial sectors. Besides, several studies were conducted on CO_2_ emissions or CO_2_ efficiency in six energy-intensive industries. As reported by Yang et al. [30], from 2007 to 2018, the carbon emissions of China’s energy-intensive industries were primarily distributed in the eastern coastal areas of China, and spatial correlation characteristics were found. Using the provincial data from 2005 to 2017, Zhu et al. [31] reported that the carbon emission efficiency of China’s energy-intensive industries had significant spatial heterogeneity and spatial agglomeration, with relatively high efficiency in eastern provinces and lower in western regions.

The influencing mechanism behind carbon emission (intensity) spatiotemporal differences is complex and is caused by a variety of drivers. The factors studied by the existing research mainly include energy structure, industrial structure, economic output, population size, energy intensity, rate of urbanization, technological advancement, foreign trade, etc. Most studies show that economic growth will lead to increased carbon emissions [32,33]. On the one hand, the increase in residents’ income will prompt residents to live a high-energy-consuming lifestyle, resulting in increased carbon emissions; on the other hand, it will also prompt residents to realize the limitations of resources and the environment and make efforts to adopt energy-saving technologies to reduce carbon emission intensity [34]. Urbanization has maintained a positive correlation with carbon emissions for a long time and is an essential factor in promoting China’s carbon emissions [35]. However, there is a certain debate about the direction of the effect of economic level and urbanization rate on carbon intensity. Some scholars believe that both of them have a negative correlation with carbon intensity, while some scholars believe that the two have a non-linear effect in addition to a direct linear effect on carbon intensity [36]. Technological progress is considered to be one of the most important factors in reducing carbon intensity. Typical domestic and foreign technologies include the reduction of fuel combustion in the process, processing conversion, and end use, as well as technologies such as carbon recovery, low-carbon energy, and carbon sinks [37,38,39]. Scholars generally believe that the industrial structure and energy structure contribute most to China’s carbon intensity effect, and that structural adjustment is still the main direction of China’s future emission reduction [40]. In addition, energy intensity is the main factor driving the reduction of China’s carbon intensity [41]. The impact of foreign direct investment on the host country’s carbon intensity is twofold; carbon intensity can either be increased through the transfer of pollution-intensive industries or reduced by technology spillovers that make related production processes cleaner [42]. Guan et al. [43], Xu et al. [44], Ouyang and Lin [27], and other studies have shown that industrial activity is a key factor in increasing CO_2_ emissions. Xie et al. [45] found that energy-intensive industries such as power production, petroleum processing, coking, chemical products, metal smelting and rolling, and non-metallic mineral products have the most prominent impact on carbon emissions. For the above energy-intensive industries, Lin and Long [46] argue that energy intensity and energy mix may be beneficial for reducing CO_2_ emissions from China’s chemical industry. Yang et al. [30] found that growth in GDP and population per capita increases CO_2_ emissions from energy-intensive industries, while the share of service industries leads to a reduction in CO_2_ emissions. Scholars have used various methods to analyze the drivers of carbon intensity under different research backgrounds, which mainly included Index decomposition analysis (IDA) [47,48], Structural decomposition analysis (SDA) [49,50], and econometric regression analysis based on the IPAT model or STIRPAT model [51,52].

The extensive studies have provided a very useful reference for the text, whereas the research on energy-intensive industries still has deficiencies. First, existing studies lack the exploration of the spatial differences in the carbon intensity of energy-intensive industries, so it is difficult to provide help for the formulation of scientific regional decomposition plans in the process of emission reduction in this key industry. Second, spatial elements often have spillover effects and the first law of geography also highlights that the closer the spatial distance between things, the stronger the interdependence will be [53]. Most of the existing research has recognized regions as separate individuals, and the spatial dependence and spatial heterogeneity of carbon intensity have been rarely discussed from the perspective of geography and spatial interaction. In this way, it is difficult to reveal the spatial correlation and provide a scientific basis for proposing emission reduction measures for joint prevention and control. To fill this research gap, this study first finds the characteristics of spatiotemporal differences in EIICI and then uses the spatial econometric method based on the STIRPAT model to investigate the drivers and spatial spillover effects of EIICI.

## 3. Materials and Methods

This section may be divided into subheadings. It should provide a concise and precise description of the experimental results, their interpretation, as well as the experimental conclusions that can be drawn.

### 3.1. Measurement of EIICI

Regarding the selection of indicators of carbon intensity, most of the existing studies use the ratio of carbon emissions to economic output, that is, the carbon dioxide emitted per unit of economic output [5,24,26,52]. We draw on similar expressions. Based on the existing research results on CO_2_ of energy-intensive industries [54,55], the EIICI is expressed as the amount of CO_2_ emitted per unit of industrial sales output value. The specific calculation formula is written as:(1)$$\mathrm{EIICI}=\frac{\sum_{i=1}^{6}CE_{i}}{\sum_{i=1}^{6}{\mathrm{Output}}_{i}}$$
where EIICI denotes the carbon intensity of energy-intensive industries; $CE_{i}$ is the CO_2_ emission of the *i* industry; ${\mathrm{Output}}_{i}$ is the industrial sales output value of the *i* industry.

### 3.2. Spatial Differences and Spatial Autocorrelation

#### 3.2.1. Coefficient of Variation

The coefficient of variation (CV) is adopted to measure the degree of regional differences in EIICI and to express the spatiotemporal dynamic differences of EIICI in China. This coefficient has been extensively used to study spatial differences in geographic data [56]. The calculation formula is written as:(2)$$C_{v}=\frac{1}{\bar{Y}}\sqrt{\frac{\sum_{i=1}^{n}{\left(Y_{i}-\bar{Y}\right)}^{2}}{n\;-\;1}}$$
where $C_{v}$ denotes the coefficient of variation; $\bar{Y}$ is the average number of EIICI in each province; *n* is the number of provinces; $Y_{i}$ represents the EIICI in each province. The larger the value of $C_{v}$, the larger the relative difference of EIICI between provinces and regions will be.

#### 3.2.2. Spatial Autocorrelation

Spatial autocorrelation analysis can examine whether there is a spatial relationship between the carbon intensity of regional industries, and it is also a necessary condition for constructing a spatial econometric model correctly. This study uses the global spatial autocorrelation test index Global Moran’s *I* to measure the degree of spatial agglomeration of EIICI [57].
(3)Moran’s I=∑i=1n∑j=1nwij(Yi−Y¯)(Yj−Y¯)∑i=1n(Yi−Y¯)2n*1∑i=1n∑j=1nwij
where $\bar{Y}\;$ is the mean value of all *n* regional observations; $w_{ij}$ is the spatial weight matrix, this study uses the First Order Contiguity Matrix. To be specific, when provinces *i* and *j* are adjacent, the value of $w_{ij}\;$ is 1; otherwise, the value is 0. $Y_{i}$ and $Y_{j}$ represent observations at spatial positions *i* and *j*. Consistent with the correlation coefficient, Moran’s *I* has a value interval of [−1, 1].

### 3.3. Spatial Econometric Model

Common spatial econometric models include the spatial lag model (SLM), the spatial error model (SEM), and the spatial Durbin model (SDM). The SLM model includes endogenous interaction effects, the spatial error model consists of the interaction effects between error terms, and the SDM model refers to a more general form of econometric model combining the characteristics of the SLM model and the SEM model [52]. The basic expression of the SDM model is written as:(4)$$Y=\rho WY+X\beta +WX\theta +\varepsilon \;;\;\varepsilon \sim N\left(0,\;\delta^{2}\right)$$
where *Y* represents the explained variable, which is a vector of n × 1; *X* denotes the explanatory variable, and if there are m explained variables, it turns out to be a matrix of (n × m); *β* represents the regression coefficient, which is a vector (m × 1); *ε* is the random error term; *N* is the number of spatial units; *W* is the spatial weight matrix (n × n); *ρ* is the autoregressive spatial coefficient. When the *ρ* value is significant, there is a certain spatial correlation between the explained variables; *γ* is the spatial correlation coefficient between the regression residuals; U is the random error vector; *θ* is the exogenous interaction effect coefficient. If H0: *θ* = 0 is established, the SDM model degenerates into the SLM model. If H0: *θ* + *ρβ* = 0, it degenerates into the SEM model; otherwise, it turns out to be the SDM model [51]. Notably, to improve the robustness of the conclusions, three spatial matrices are compared. The first is the binary rook contiguity spatial weights matrix (W1). QUEEN adjacency is adopted to assign values to elements in the spatial weight matrix in accordance with whether they are geographically adjacent. The value is 1 if the space unit is adjacent; otherwise, the value is 0. The second is the inverse distance-based spatial weights matrix (W2), where the elements of the matrix are assigned the inverse of the straight-line distance between spatial units. The third refers to the economic weight matrix (W3). The value of the elements in the matrix is the inverse of the absolute value of the difference in GDP per capita between the two places, which reveals the economic similarity between the two places.

When the spatial model is applied specifically, which spatial panel model to choose and which fixed effect should be included should be determined step by step during testing and diagnosis [58]. Since some scholars have highlighted that using the point estimation method to test spatial spillover effects would cause errors, they proposed to measure the direct effects and indirect effects between adjacent regions according to partial differential. Elhorst [59] extended the above method to the spatial panel model. The partial differential matrix ${\left(I-\rho W\right)}^{-1}\left(\beta_{k}+W\theta_{k}\right)$ is obtained by calculating the partial derivative, the average value of the main diagonal elements of the matrix refers to the direct effect, while the mean of elements other than the main diagonal is the indirect effect.

### 3.4. Model Specification

The IPAT (Impact, Population, Affluence, Technology) model was originally proposed by Ehrlich and Holdren [60] to investigate changes in environmental stress driven by human activities. During the practical application of the IPAT model, it has been continuously supplemented and expanded by scholars, and the STIRPAT model has been derived. The STIRPAT model is a stochastic representation of the IPAT equation and a multivariate nonlinear model. After logarithmic processing, multivariate linear fitting can be performed, which realizes the causal relationship analysis of environmental impacts by adding different human factors to the model [61]. This study takes the STIRPAT model as the basic framework to analyze the drivers of EIICI.

The drivers affecting EIICI consist of numerous aspects. According to the above literature review and summary and the availability of data, this study primarily investigates the following factors (economic level, urbanization level, technological innovation, energy structure, environmental regulation, industrial agglomeration, as well as firm size). In this study, the level of economic development, the urbanization level, the technological innovation, and the energy structure are expressed by per capita GDP, the proportion of urban population to the total population, per capita R&D expenditure, and the proportion of coal in energy consumption, respectively. Besides, the environmental regulation, the industrial agglomeration, and the firm size are expressed by the investment in air pollution control per unit of emission [62], the share of the total industrial output value of the energy-intensive industry in all industries, and the average total industrial output value of energy-intensive enterprises, respectively.

Based on the STIRPAT model, the EIICI serves as the explained variable, and the above seven drivers are included in the model as the explanatory variables. Lastly, the general panel data regression model is adopted to explain the drivers of China’s EIICI. On that basis, further spatial econometric analysis is conducted. To make the data more in line with the normal distribution and eliminate the heteroscedasticity of the model, logarithmic transformation is performed on the explained variables and explanatory variables before regression.
(5)$$Ln{\mathrm{EIICI}}_{it}=\alpha_{it}+\beta_{1}Ln{\mathrm{ECON}}_{it}+\beta_{2}Ln{\mathrm{URB}}_{it}+\beta_{3}LnINN_{it}+\beta_{4}LnES_{it}+\beta_{5}Ln{\mathrm{ER}}_{it}\;+\beta_{6}Ln{\mathrm{AGG}}_{it}+\beta_{7}Ln{\mathrm{FZ}}_{it}+\varepsilon_{it}$$
where Ln represents the natural logarithm; *i* (*i* = 1, 2, …, 30) is 30 provinces; *t* denotes the time interval (*t* = 1, 2, …, 20); ECON means regional economic level; URB is urbanization level; INN expresses technological innovation; ES denotes energy structure; ER is environmental regulation; AGG is industrial agglomeration; FZ is firm size; $a_{it}$ expresses that there are different fixed effects in the cointegration relationship of each panel unit; $\beta_{i}$ denotes the elastic coefficient of each explanatory variable; $\varepsilon_{it}$ is the random error term of the cross-sectional individual *i* at time *t*.

### 3.5. Data Source and Description

The criteria for the definition of energy-intensive industries in the “National Economic and Social Development Statistical Bulletin of China in 2019” are adopted in this study. The standard follows the national economic industry classification method, completely consistent with the industrial statistics published by the National Bureau of Statistics. The six major industries covered in the energy-intensive industry and their industry codes in the “National Economic Industry Classification (2011)” consist of petroleum refining and coking (25), chemical raw material and chemical product manufacturing (26), non-metallic mineral products (30), iron and steel (31), non-ferrous metals (32), as well as the power industry (44). Data on carbon emissions of energy-intensive industries by province over the years can be downloaded freely from Carbon Emission Accounts and Data sets for emerging countries (CEADs) (https://www.ceads.net/data/province/, accessed on 4 December 2021). The total industrial output value of the energy-intensive industry originates from the “China Industrial Economic Statistical Yearbook” over the years, the consumption data of each energy by industry originate from the “China Energy Statistical Yearbook”, and the data of the remaining explanatory variables, such as GDP, population, FDI, R&D expenditure, number of enterprises, etc., come from the “China Statistical Yearbook”, “China Science and Technology Statistical Yearbook”, “China Environmental Yearbook”, “China Trade and Foreign Economic Statistical Yearbook” as well as statistical yearbooks of different provinces (https://tongji.oversea.cnki.net/oversea/engnavi/navidefault.aspx, accessed on 21 December 2021). Table 1 lists the basic statistical descriptions of the explanatory variables and the explained variables obtained. The time span of the basic data employed in this study is between 2000 and 2019, and the basic spatial unit is 30 provinces (Hong Kong, Macao, Taiwan and Tibet are not included due to missing data). Map vector data originate from the National Geomatics Center of China (http://www.ngcc.cn/, accessed on 1 December 2021).

## 4. Spatiotemporal Differences of EIICI

(1)From 2000 to 2019, the EIICI showed a significant phased downward trend

The EIICI in 2000 reached up to 8.70 t/10^4^ yuan since China was in the middle stage of industrialization around the year 2000. At that time, China’s economic growth was largely dependent on large-scale consumption of resources and the environment, the level of technical management was relatively low, and the development method was relatively extensive, thus leading to large-scale energy consumption and CO_2_ emissions. As the economic level and the low-carbon development path have been continuously improved, the EIICI has been reducing due to several factors (e.g., industrial upgrading, technological progress, as well as environmental regulation). According to the rate of descent, there are two distinct stages. The first stage is between 2000 and 2009, during which the EIICI decreased rapidly, with an average annual decrease of 0.54 t/10^4^ yuan. By 2010, the EIICI quickly declined to 3.32 t/10^4^ yuan. The second stage is from 2010 to 2019, with an average annual decrease of 0.12 t/10^4^ yuan. By 2019, its intensity will be adjusted to 2.13 t/10^4^ yuan, nearly 1/4 of that in 2000.

(2)At the provincial scale, the EIICI shows a significant downward trend, but there are gradually expanding spatial differences

In general, China can be divided into three major regions (Table 2). As indicated by the calculation results of the coefficient of variation (CV) from 2000 to 2019, the CV of EIICI has progressively increased from 0.42 to 0.64, which implies that the regional gap tends to expand (Figure 1). To describe the relative differences in EIICI between regions, this study uses the World Bank’s classification method for regional economic development levels [63] and adopts 50%, 100%, and 150% of the average EIICI of each province in each year as node values. It is divided into four types: high intensity, medium-high intensity, medium-low intensity, and low intensity. In 2000, there were six high-intensity type areas, including Shaanxi, Guizhou, Anhui, Shanxi, Chongqing, and Inner Mongolia, all of which are located in the central and western regions, and the overall spatial distribution was banded. Besides, there were five medium-high intensity types, including Jilin, Hebei, Heilongjiang, Guangxi, and Ningxia, all of which are located in the central and western regions except for Hebei. The number of low-intensity type areas was relatively small, with only the five areas of Beijing, Tianjin, Shanghai, Guangdong, as well as Xinjiang. Except for Xinjiang, the other four are coastal provinces with developed economies. The other 14 provinces were the low-medium intensity type areas, characterized by a contiguous distribution in space. In 2010, the EIICI of Jiangsu, Zhejiang, Hebei, Anhui, and Chongqing was reduced, while the EIICI of Yunnan, Henan, Ningxia, Xinjiang, and Hainan increased. To be specific, Xinjiang and Hainan shifted the most. Xinjiang has shifted from a low-intensity type to a medium-high intensity type, and Hainan has shifted from a medium-low intensity type to a high-intensity type. In 2019, the spatial distribution pattern of provinces and regions has changed, and the number of four types, including high intensity, medium-high intensity, medium-low intensity, and low-intensity provinces, has been adjusted to 6, 7, 10, and 7, respectively. The high-intensity type areas all belong to the northern provinces, and the southern provinces are mostly two types of low-intensity and low-intensity areas (Figure 2).

(3)At the regional scale, the EIICI shows a pattern of regional differences in the coexistence of “high in the west and low in the east” and “high in the north and low in the south”

For the differences between the three major regions in China, the CV of EIICI in the eastern, central, and western regions from 2000 to 2019 increased from 0.29, 0.25, and 0.39 to 0.46, 0.55, and 0.57, respectively, thus revealing that the regional differences in China’s EIICI have formed a pattern in which the west was larger than the central region and the eastern region, and the differences within the region were also expanding. In 2000, the EIICI of the eastern, central, and western regions reached 7.05 t/10^4^ yuan, 11.99 t/10^4^ yuan, and 13.20 t/10^4^ yuan, respectively. In 2019, the EIICI of the eastern, central, and western regions was regulated to 1.57 t/10^4^ yuan, 2.95 t/10^4^ yuan, and 3.15 t/10^4^ yuan, respectively, and the central and western regions were 1.88 times and 2.01 times higher than the eastern region, respectively. Furthermore, from the perspective of the differences between the north and the south, ten northern provinces of Xinjiang, Gansu, Inner Mongolia, Ningxia, Shaanxi, Shanxi, Hebei, Liaoning, Jilin, and Heilongjiang are taken in this study as a whole to examine the difference in EIICI between the northern provinces and the rest of the southern provinces. In 2000, the average EIICI of northern provinces and other provinces was 15.73 t/10^4^ yuan and 7.70 t/10^4^ yuan, respectively, the former was 2.04 times that of the latter. In 2019, the average EIICI of the two was regulated to 4.24 t/10^4^ yuan and 1.66 t/10^4^ yuan, respectively, and the gap between the former and the latter increased to 2.55 times. Thus, China’s EIICI shows significant “high in the west and low in the east” and “high in the north and low in the south” patterns, while the “difference between the north and the south” is higher than the “difference between the east and the west”.

(4)EIICI has significant positive spatial autocorrelation

The global Moran’s I index is adopted to measure the degree of spatial autocorrelation of EIICI from 2000 to 2019, and the random permutation method is employed to construct a normal distribution to examine its significance (Table 3). As indicated by the results, the global Moran’s I in each year is all positive, and the statistic Z index is all significant at the 1% level (*p* values all less than 0.01), thus revealing a significant positive spatial autocorrelation of EIICI in China, i.e., the EIICI of this unit positively impacts the adjacent units. In contrast, the adjacent units will have a positive effect on this unit as well. From the perspective of time evolution, the Moran’s I from 2000 to 2019 showed a fluctuating upward trend, increasing from 0.282 to 0.377, indicating that the agglomeration degree of China’s EIICI has increased, and the types of areas with similar EIICI are more spatially inclined in agglomeration distribution. Thus, spatial effects should be added to improve the accuracy of model estimation in the panel data regression model adopted to explore the drivers below. Furthermore, the local spatial agglomeration characteristics of EIICI are significant. The provinces with high-high agglomeration are largely distributed in Guangxi–Guizhou and northern China, while the provinces exhibiting low-low agglomeration are concentrated in the eastern coastal zone.

## 5. Estimation Results and Discussion of Drivers

### 5.1. Model Test

In this study, 20 years of data with a long time span are involved, so the stationarity of panel data should be analyzed. As indicated by the unit root test, the panel data of EIICI from 2000 to 2019 is stable, and the logarithm values of the explanatory variables are stable after the first-order difference. As indicated by the cointegration test results, there is a cointegration relationship between the above variables. Besides, there is a significant spatial autocorrelation in the explained variable EIICI; at the same time, the standard error estimated by the OLS method for the ordinary panel regression model is investigated by spatial autocorrelation, and the global Moran’s I index is 2.626, significant at 1% level. Accordingly, for non-independent sample data, the results estimated using conventional OLS methods may have serious biases. Spatial effects should be incorporated into the model, and a more suitable spatial panel regression model should be used.

As revealed by the results of the Hausman test (26.13, *p* = 0.001), the general panel data regression model should be built in a space-fixed form. Further, the LM test and robustness LM test are performed on the model without spatial interaction, which consist of four types of spatial and temporal non-fixed (mixed), spatially fixed, temporally fixed, and temporally fixed, to verify whether SEM and SLM outperform non-spatial models. The adjacency weight matrix (W1) is selected as the initial spatial weight matrix. Table 4 lists the results of the LM test and robust LM test for the four forms. Except that the spatial lag effect in the mixed form and the time-fixed form do not pass the robust LM test at the 10% level, the spatial lag effect and the spatial error effect in the spatial fixed form and the two-way fixed form pass the LM test and the robust LM test. Furthermore, according to the goodness-of-fit test statistic and log-L value, the spatiotemporal fixed effects model is verified to be significantly better than the other three models. Thus, a more general form of the SDM model should be further constructed, and the model should be tested in the fixed form of space and time.

In addition, as revealed by the Wald test and LR test results of the SDM model, the four statistics of Wald-lag, Wald-error, LR-lag, and LR-error all pass the 1% significance level. Accordingly, the SDM model cannot degenerate into the SEM or SLM model. To improve the robustness of the conclusions, the SDM is also calculated under the inverse distance weight matrix (W2) and economic weight matrix (W3), respectively, and the results are listed in Table 5. By comparing the adjusted R^2^ and Log Likelihood of SDM under W1, W2, and W3, the SDM simulation effect under W2 is found as the best. Thus, the SDM model with two-way fixed effects in time and space under W2 is finally selected to reveal the effect arising from various factors on the spatiotemporal characteristics of China’s EIICI.

### 5.2. Analysis of Estimation Results

#### 5.2.1. Results of SDM Model

The EIICI has been reported to exert significant spatial spillover effects. In the SDM model under the W2 matrix, the value of the spatial lag term ρ is obtained as 0.484, positive and significant at a 1% level, which reveals that local EIICI is significantly affected by neighboring provinces while affecting the EIICI of neighboring provinces. For each 1% change in EIICI of neighboring provinces, the local EIICI will change by 0.484% in the same direction. Some studies have found similar spatial spillover effects when studying the emissions of air pollutants and water pollutants [57,64]. This spatial spillover effect of EIICI can be largely explained below. First, due to the geographical proximity of natural resource distribution, adjacent areas generally have similar industrial raw materials (e.g., coal and iron ore) [65], thus laying a natural basis for similar industrial structures. Moreover, adjacent areas are generally in the same regional economic division system, while having commonalities in production costs, markets, as well as technologies. If the local energy-intensive industry is difficult to clean, the emission reduction of adjacent areas is also in a disadvantageous situation. Second, the spatial spillover effect arises from the proximity transfer of energy-intensive industries. Although local governments have implemented more rigorous environmental control policies over the past few years, some energy-intensive industries have been relocated to cut environmental governance costs. However, due to the combination of factors (e.g., the proximity of raw materials, consumer preferences, and transportation costs), even if the energy-intensive industries are relocated, a large-scale spatial transfer cannot be achieved to distant areas.

The elasticity coefficient of LnECON to EIICI is −0.397, which is statistically significant at a 1% level, indicating that the economic level has a promoting effect on the reduction of the EIICI. For each 1% increase in economic level, the EIICI will decrease by 0.397%, consistent with the findings of other scholars on overall carbon intensity. The elasticity coefficient of technological innovation is negative and significant at a 1% level, thus indicating that the stronger the technological innovation capability, the more significant the corresponding reduction of CO_2_ emissions from energy-intensive enterprises will be. In addition, the improvement of technological level can boost the improvement of energy utilization efficiency and the development and utilization of new energy sources, thus helping reduce CO_2_ emissions in production and reduce EIICI. The elastic coefficients of LnAGG and LnFZ are −0.172 and −0.125, respectively, both of which are statistically significant at a 1% level, indicating that both the industrial agglomeration and the firm size are conducive to reducing the EIICI. There is a scale effect at the enterprise level. The larger the firm size, the more the pollutants should be treated, and the lower the marginal cost of purchasing pollutant treatment facilities will be. Thus, companies are inclined to spend more money on efficient equipment [66]. A considerable number of energy-intensive enterprises are located in a single industrial park or industrial parks with relatively short distances, often forming energy-intensive industry agglomeration [67]. Industrial parks are capable of planning for polluting facilities and constructing the above facilities in a unified manner, thus having agglomeration effects on pollution reduction, which can reduce the pollution investment cost of a single enterprise while increasing the efficiency of pollutant treatment. The elasticity coefficient of LnES to EIICI is 0.363, which is significant at a 1% level, indicating that the coal-dominated energy structure is a vital driver hindering the decline of EIICI. Each 1% increase in the proportion of coal consumption will lead to an increase of 0.363% in EIICI. At present, traditional fossil energy still dominates the energy consumption of the energy-intensive industries, and there is still room for improvement in energy efficiency. Furthermore, the elastic coefficients of urbanization and environmental regulation on EIICI are negative and positive, respectively, whereas the above elastic coefficients are not statistically significant at the 10% level.

#### 5.2.2. Direct and Indirect Effects

The spatial relationship of variables in the model can be further determined according to direct effects, indirect effects, as well as total effects [58]. The direct effect refers to the effect arising from an explanatory variable in a region on the explained variable in this region, which covers the effect of the explanatory variable on the explained variable in other regions after it affects the explained variable in this region, i.e., the feedback effect. Indirect effects represent the effect arising from the explanatory variables in the neighboring regions on the local explained variables. The total effect refers to the sum of the two effects, representing the average effect of the explanatory variable in a region on the explained variable in all regions. Table 6 shows that drivers can fall into four types in accordance with whether the direct and indirect effects are significant.

The first type of driver has direct and indirect effects simultaneously, which consists of technological innovation, energy structure, and industrial agglomeration. The coefficients of the direct effect and indirect effect of technological innovation can be both negative and significant, and the indirect effect is higher than the direct effect, which implies that technological innovation plays a role in reducing the EIICI, and the effect of spatial spillover effects is more significant. The level of technological innovation increases by 1% on average and the EIICI of local and adjacent regions is reduced by 0.092% and 0.508%, respectively. Thus, the effect arising from technological innovation on the reduction of EIICI shows a wide spatial range. The coefficients of the direct effect and indirect effect of the energy structure can be both negative and significant, and the indirect effect is smaller than the direct effect. For each 1% increase in the proportion of coal in energy consumption, the EIICI of local and adjacent regions will increase by 0.320% and 0.090%, respectively. Coal resources have been concentrated and contiguously distributed in China’s northern region, and the price of coal is lower than that of other energy sources. Thus, energy-intensive enterprises purchase more coal resources from local places or neighboring regions. The direct effect of industrial agglomeration is negative, and the indirect effect arising from industrial agglomeration is positive, whereas the direct effect is significantly larger than the indirect effect. Each 1% increase in industrial agglomeration reduces the local EIICI by 0.176% while increasing the EIICI in adjacent regions by 0.080%. This special phenomenon may be explained in that during the process of agglomeration of industries to a province, the industrial agglomeration of adjacent provinces will be reduced. Consequently, neighboring provinces lose their industrial agglomeration effect, thus leading to an increase in the EIICI.

The second type of driver only has direct effects and no indirect effects, which consists of economic level and firm size. For each 1% increase in economic level and firm size, the local EIICI is reduced by 0.397% and 0.126%, respectively, whereas it does not affect adjacent regions. The third type of driver has only indirect effects and no direct effects, and only environmental regulation belongs to this category. The coefficient of the indirect effect of environmental regulation is 0.033, significant at a 5% level, which indicates that a 1% increase in environmental regulation intensity has no significant effect on local EIICI, but will lead to an increase in the EIICI of surrounding provinces by 0.033%. After the local environmental control is optimized, energy-intensive enterprises face increased pollution control costs. Instead of investing in advanced technology and environmental protection equipment to increase cleaning efficiency, some companies have been relocated to neighboring provinces with looser environmental regulations [57]. As a result, the local EIICI has not been significantly reduced, whereas the surrounding provinces have increased the c EIICI because of the increase of low-tech energy-intensive enterprises. The fourth driver has neither direct effect nor indirect effect, only the urbanization level belongs to this category. The direct effect of urbanization was negative and the indirect effect was positive, but neither passed the significance test at the 10% level. In response to this finding, it is not consistent with the research results of existing studies on the overall carbon intensity, which suggested that urbanization levels significantly reduced carbon intensity [32]. It is therefore revealed that there are differences in the influencing mechanisms of each driver on the EIICI and overall carbon intensity, and more targeted measures should be formulated to reduce the EIICI.

## 6. Conclusions

Reducing the EIICI has been a vital starting point for China to achieve its carbon emission reduction goals by 2030. Based on the panel data of 30 provinces in China from 2000 to 2019, this study primarily investigates the spatiotemporal differences in EIICI and its drivers. The main conclusions are drawn below. First, the downward trend of China’s EIICI was extremely significant from 8.70 t/10^4^ yuan to 2.13 t/10^4^ yuan between 2000 and 2019. Second, the EIICI exhibits significant spatial differentiation characteristics, and the spatial differences at both provincial and regional scales tend to expand. It shows a spatial pattern of coexistence of “high in the west and low in the east” and “high in the north and low in the south”, and the “north-south gap” is larger than the “east-west gap”. Third, the EIICI has significant spatial spillover effects and the local EIICI is obviously affected by the surrounding areas. For each 1% change in the EIICI of the adjacent province, the local province will change by 0.484% in the same direction. Fourth, for drivers, technological innovation not only reduces the local EIICI, but negatively affects the adjacent regions by exploiting spillover effects. The proportion of coal in energy consumption significantly hinders the reduction of EIICI in both local and adjacent regions. Industrial agglomeration reduces the local EIICI, while it positively impacts the adjacent regions. Economic levels and firm sizes only have direct effects instead of indirect effects, and both of which have negative effects on the local EIICI. Environmental regulation only has spatial spillover effects instead of direct effects. If the intensity of environmental regulation increases by 1%, the EIICI of adjacent regions will increase by 0.033%; the urbanization level has neither direct nor indirect effects.

## 7. Implications and Future Study

The above results have high implications for policy. First, the dominant drivers affecting the EIICI should be determined, as well as the key direction of emission reduction policies. They emphasize the role of technological innovation in reducing EIICI and combine independent innovation with the introduction of advanced foreign industrial technology and management experience. In order to adjust the energy consumption structure of high-energy-consuming industries and increase the share of clean energy consumption, the government needs to provide more incentives (such as increasing environmental taxes and increasing clean energy subsidies). Given the characteristics of most high-energy-consuming enterprises with small scale and scattered locations, we should fully exploit the scale effect, optimize the industrial space layout and facilitate the agglomerated development of energy-intensive industries. It is necessary to strengthen environmental control, formulate strict emission standards, and appropriately increase punitive damages for companies exceeding the standards. Second, targeted emission reduction measures should be formulated based on the reality of regional differences. In the eastern coastal areas, independent innovation and the introduction of foreign advanced technologies should be facilitated, the application of innovative technologies in industrial production should be boosted, and ecological industrial clusters and a circular economy should be created. The central and western regions and northern provinces are required to largely optimize the energy consumption structure and progressively reduce the high consumption and high emission development mode dominated by coal. In the meantime, the transfer of energy-intensive industries from the eastern region should be actively and effectively dealt with to avoid becoming a “pollution heaven”. Local governments in the central and western regions should strengthen environmental supervision and rationally facilitate industrial layout under the premise that the carrying capacity of resources and the environment is evaluated. Third, stress should be placed on the spatial spillover effect of EIICI, and regional coordinated development and control strategies should be formulated. In particular, when formulating economic development policies, the governments should consider the transfer of carbon emissions to emphasize the inhibition of local EIICI, as well as to consider the effect on surrounding areas. To reduce the overall carbon intensity of the region, administrative boundaries should be broken, and a cooperative governance model of overall planning, resource sharing, industrial collaboration, and information sharing should be implemented in adjacent regions.

This study still has some limitations, and some issues should be explored in depth. First, it can be analyzed according to smaller spatial scales and basic units. Impacted by data limitations, only provincial data are selected in this study. To improve the precision and accuracy of the research, further research using prefecture-level cities or counties as the basic spatial unit should be conducted. Second, stress should be placed on the research on the sub-sectors of energy-intensive industries. Since there are significant differences in the factors that determine the carbon intensity of various industries, in-depth discussions should be conducted from the sub-sectors of energy-intensive industries in the future, which is conducive to improving the pertinence of energy conservation and emission reduction in different regions. Third, new models and novel methods should be adopted to analyze the influencing mechanism of key drivers and the interaction between multiple drivers. This study primarily aims to reveal the influence direction, relative strength, and spillover effect of different factors. Future research needs to further consider the bilateral causal relationship between the dependent and independent variables to address the endogeneity problem. In the study of long-term series, it is necessary to further analyze the influence of factors such as economic purchases on the results. A geospatial weighted regression model should be used to explore the impact intensity of different factors in each province, in order to reveal the differences in the driving mechanisms between regions. Finally, by conducting a regional comparative study, it is investigated whether the relevant calculation results can be verified in typical case areas.

## Figures and Tables

**Figure 1 ijerph-19-10235-f001:**
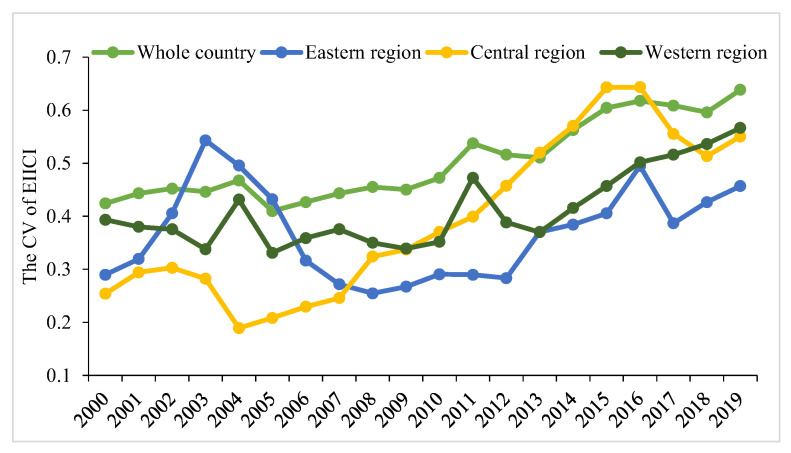
Variation coefficient of EIICI across the country and regions from 2000 to 2019.

**Figure 2 ijerph-19-10235-f002:**
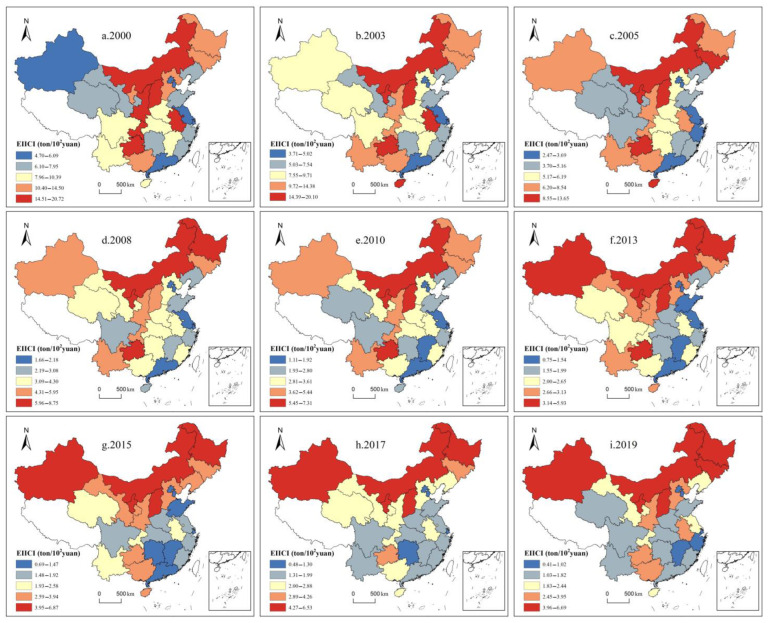
Spatiotemporal differences of China’s EIICI from 2000 to 2019.

**Table 1 ijerph-19-10235-t001:** Description of the variables.

Variable	Symbol	Unit	Min	Mean	Max	SD
Carbon intensity of energy-intensive industries	EIICI	Tons/10^4^ yuan	0.410	5.100	23.83	4.010
Economic level	ECON	Yuan	2759	34,702	160,000	27,385
Urbanization	URB	%	20.35	51.06	89.60	15.02
Technological innovation	INN	Yuan/person	10.53	686.5	11,256	1187
Energy structure	ES	%	1.210	47.19	92.64	17.84
Environmental regulation	ER	10^4^ yuan/tons	0.020	1.700	52.09	5.310
Industrial agglomeration	AGG	%	18.06	39.09	75.77	13.10
Firm size	FZ	10^4^ yuan	797.5	8071	31,533	6466

**Table 2 ijerph-19-10235-t002:** Distribution of the three major regions in China.

Regions	Provinces
Eastern region	Beijing, Tianjin, Hebei, Liaoning, Shanghai, Jiangsu, Zhejiang, Fujian, Shandong, Guangdong, Hainan
Central region	Shanxi, Jilin, Heilongjiang, Anhui, Jiangxi, Henan, Hubei, Hunan
Western region	Inner Mongolia, Guangxi, Chongqing, Sichuan, Guizhou, Yunnan, Shannxi, Gansu, Qinghai, Ningxia, Xinjiang

**Table 3 ijerph-19-10235-t003:** Global Moran’s I index and Z statistics of China’s EIICI from 2000 to 2019.

Year	2000	2001	2002	2003	2004	2005	2006	2007	2008	2009
Moran’I	0.282	0.304	0.389	0.213	0.270	0.345	0.294	0.303	0.355	0.312
Z statistics	2.772	2.964	3.746	2.185	2.728	3.355	2.955	3.022	3.482	3.066
*p* value	0.006	0.003	0.000	0.029	0.006	0.001	0.003	0.003	0.000	0.002
**Year**	**2010**	**2011**	**2012**	**2013**	**2014**	**2015**	**2016**	**2017**	**2018**	**2019**
Moran’I	0.363	0.275	0.323	0.312	0.345	0.332	0.366	0.338	0.366	0.377
Z statistics	3.505	2.842	3.169	3.059	3.369	3.265	3.581	3.306	3.562	3.649
*p* value	0.000	0.004	0.002	0.002	0.001	0.001	0.000	0.001	0.000	0.000

**Table 4 ijerph-19-10235-t004:** LM test results of four forms under the common panel model.

Statistics	Mixed Effects	Spatial Fixed Effects	Time-Period Fixed Effects	Two-Way Fixed Effects
R^2^	0.747	0.825	0.767	0.863
Log-L	−201.402	−191.128	−44.622	−20.799
LM-lag	107.829 (0.000) ***	201.835 (0.000) ***	51.732 (0.000) ***	51.866 (0.000) ***
Robust LM-lag	1.358(0.561)	3.513 (0.086) *	2.741 (0.107)	4.948 (0.015) **
LM-error	608.595 (0.000) ***	644.876 (0.000) ***	326.733 (0.000) ***	353.689 (0.000) ***
Robust LM-error	501.123 (0.000) ***	445.782 (0.000) ***	275.515 (0.000) ***	304.771 (0.000) ***

Note: *, **, and ***, respectively, denote significance at different levels (10%, 5% and 1%).

**Table 5 ijerph-19-10235-t005:** Estimation results of SDM model under three spatial matrix conditions.

	W1	W2	W3
LnECON	−0.390 *** (−5.41)	−0.397 *** (−5.98)	−0.443 *** (−5.63)
LnURB	−0.120 (−1.07)	−0.0383 (−0.37)	−0.355 ** (−2.75)
LnINN	−0.0839 ** (−2.97)	−0.0947 *** (−3.51)	−0.132 *** (−4.41)
LnES	0.375 *** (11.53)	0.363 *** (11.44)	0.310 *** (7.90)
LnER	0.0164 (0.96)	0.0195 (1.25)	0.0278 (1.54)
LnAGG	−0.117 (−1.96)	−0.172 *** (−3.31)	0.0129 (0.20)
LnFZ	−0.191 *** (−5.48)	−0.125 *** (−3.98)	−0.122 ** (−3.15)
*ρ*	0.249 *** (4.35)	0.484 *** (10.80)	0.285 *** (5.32)
W*LnECON	0.373 ** (2.82)	0.0972 (0.64)	0.419 (1.57)
W*LnURB	−0.389 (−1.85)	0.140 (0.56)	0.787 * (2.13)
W*LnINN	−0.318 *** (−5.12)	−0.546 *** (−8.28)	0.0407(0.46)
W*LnES	−0.0922 (−1.32)	−0.105 (−1.13)	−0.0627 (−0.69)
W*LnER	0.0411 * (2.54)	0.0479 *** (3.29)	0.0278 (1.61)
W*LnAGG	0.506 ** (3.05)	0.479 * (2.12)	1.307 *** (7.06)
W*LnFZ	−0.198 (−1.92)	−0.0595 (−0.61)	−0.291 ** (−3.22)
*σ* ^2^	0.0204 *** (17.17)	0.0169 *** (17.56)	0.0229 *** (17.31)
Adj. R^2^	0.894	0.917	0.858
Log Likelihood	312.595	361.723	281.396
N	600	600	600

Note: *, **, and ***, respectively, denote significance at different levels (10%, 5% and 1%); t statistics in parentheses.

**Table 6 ijerph-19-10235-t006:** Results of direct effects, indirect effects and total effects for SDM.

	LR_Direct	LR_Indirect	LR_Total
LnECON	−0.397 *** (−5.83)	0.0952 (0.61)	−0.301 (−1.71)
LnURB	−0.0443 (−0.43)	0.159 (0.66)	0.115 (0.45)
LnINN	−0.0915 *** (−3.56)	−0.508 *** (−7.91)	−0.599 *** (−9.11)
LnES	0.320 *** (10.87)	0.0897 ** (2.72)	0.410 *** (3.69)
LnER	0.0187 (1.20)	0.0334 ** (2.53)	0.0521 (1.73)
LnAGG	−0.176 *** (−3.34)	0.0804 * (2.35)	−0.0951 (−0.66)
LnFZ	−0.126 *** (−3.90)	−0.053 (−0.54)	−0.179 (−1.75)

Note: *, **, and ***, respectively, denote significance at different levels (10%, 5% and 1%); t statistics in parentheses.

## Data Availability

The data used to support the findings of this study are available from the corresponding author upon request.

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
