# Peer review of "What Cause Large Spatiotemporal Differences in Carbon Intensity of Energy-Intensive Industries in China? Evidence from Provincial Data during 2000–2019"

_ijerph, 2022, doi:10.3390/ijerph191610235_

Round 1
Reviewer 1 Report
To reduce the carbon intensity of energy-intensive industries, in this study, the spatiotemporal differences of EIICI are described using the panel data of 30 provinces in China from 2000 to 2019, and a spatial econometric model is further adopted to analyze its drivers. It is a carefully done study and the findings are of considerable interest. However, I thought it still has some deficiencies and I recommend to a revision before acceptable publication. Detailed comments are listed below:
Section1: Some existing works on cleaner production and environmental sustainability in the energy industry (Some developments and new insights of environmental problems and deep mining strategy for cleaner production in mines. Journal of Cleaner Production, 2019, 210: 1562-1578. Analysis of a Peaked Carbon Emission Pathway in China Toward Carbon Neutrality Project Team on the Strategy and Pathway for Peaked Carbon Emissions and Carbon Neutrality Comment. ENGINEERING, 2021, 7(12): 1673-1677.) may help to provide some useful information about achieving the two-carbon goal. Authors are encouraged to discuss these related works to better complete this manuscript and provide readers with comprehensive content.
Section1: "In 2019, China's carbon intensity was 48% lower than in 2005 and 17% below the 2030 target." mentioned in the article, is this supported by relevant data?
Section1: What is the importance of achieving the dual carbon goal? At present, my country's industrial structure adjustment is being vigorously strengthened. How does this help to achieve the dual carbon goal? Please make it clearly.
Section1: The carbon emission targets in different fields are different, and their roles are also different. How does this affect the realization of the dual carbon targets?
Section1: Some data sources in the text are not very clear, please add relevant data source descriptions or references. (For example: carbon emissions or total GDP by region).
Section2: How is carbon intensity defined? How do the development level, energy structure and industrial structure of a country affect carbon emission intensity?
Section2: It is mentioned in the manuscript that "Technological progress is considered to be one of the most important factors in reducing carbon intensity, including both domestic and foreign technologies", but what kind of technology is not clearly explained in the manuscript.
Section2: Is the formula for calculating CO2 emissions for the 6 sub-sectors justified? What are the six sub-sectors? Also, how is the emission factor corrected? Please make it clearly.
Section3: The derivation or source of some formulas in the manuscript is not very clear, please add relevant content
Section4: The amount of economic purchases in the 20-year span is different. In addition, the extensive economic development method ten years ago and the current resource utilization are also different. Will this affect the conclusion?
Section5: How can the relevant calculations in the manuscript be validated in reality? Or how to justify the conclusion?
Author Response
Dear Reviewer:
Thank you for your comments concerning our manuscript. Those comments are all valuable and very helpful for revising and improving our paper, as well as the important guiding significance to our researches. We have revised our paper carefully to address the comments and have made many corrections in the original text in revision mode, which we hope have adequately addressed your concerns. We look forward to hearing from you.
Best regards,
Authors
Comments and Responses:
To reduce the carbon intensity of energy-intensive industries, in this study, the spatiotemporal differences of EIICI are described using the panel data of 30 provinces in China from 2000 to 2019, and a spatial econometric model is further adopted to analyze its drivers. It is a carefully done study and the findings are of considerable interest. However, I thought it still has some deficiencies and I recommend to a revision before acceptable publication. Detailed comments are listed below:
Section1: Some existing works on cleaner production and environmental sustainability in the energy industry (Some developments and new insights of environmental problems and deep mining strategy for cleaner production in mines. Journal of Cleaner Production, 2019, 210: 1562-1578. Analysis of a Peaked Carbon Emission Pathway in China Toward Carbon Neutrality Project Team on the Strategy and Pathway for Peaked Carbon Emissions and Carbon Neutrality Comment. ENGINEERING, 2021, 7(12): 1673-1677.) may help to provide some useful information about achieving the two-carbon goal. Authors are encouraged to discuss these related works to better complete this manuscript and provide readers with comprehensive content.
Response:
Thank you for the important references you gave us. Focusing on the research topic of this paper, we include the key contents of the literature on cleaner production and emission reduction in key industries into the introduction. These can better prove that China must formulate effective policy measures in the field of energy-intensive industries in order to achieve the goal of reducing carbon intensity in 2030. According to your advice, we added the two references to the revised paper.
References:
- Dong, L.; Tong, X.; Li, X.; Zhou, J.; Wang, S.; Liu, B. Some developments and new insights of environmental problems and deep mining strategy for cleaner production in mines. J. Clean. Prod. 2019, 210, 1562–1578.
- Team, P.; Carbon, P.; Neutrality, C. Analysis of a Peaked Carbon Emission Pathway in China Toward Carbon Neutrality. Engineering 2021, 7, 1673–1677.
Section1: "In 2019, China's carbon intensity was 48% lower than in 2005 and 17% below the 2030 target." mentioned in the article, is this supported by relevant data?
Response:
Thank you for pointing this out. We have now cited the reference to support our point of view.
References:
- State Council Information Office of the People's Republic of China. China's energy development in the new era; People's Publishing House: Beijing, China, 2020.
Section1: What is the importance of achieving the dual carbon goal? At present, my country's industrial structure adjustment is being vigorously strengthened. How does this help to achieve the dual carbon goal? Please make it clearly.
Response:
The dual carbon goals are China's commitment to the world at the 2020 UN General Assembly general debate and the Climate Ambition Summit. The dual carbon goals will affect China's economic development mode, people's way of life, energy consumption structure and other aspects. China is making great efforts to adjust its industrial structure. Every year, it promulgates the "Industrial Structure Adjustment Guidance Catalogue", which aims to reduce the proportion of industries with high energy consumption, high emissions and high pollution. Recently, the "Implementation Guidelines for Energy Conservation and Carbon Reduction Transformation and Upgrading in Key Areas of High Energy-consuming Industries (2022 Edition)" has also been issued. Focusing on 17 industries including oil refining, cement, iron and steel, and non-ferrous metal smelting, the work directions for energy-saving and carbon-reducing transformation and upgrading and specific goals for 2025 have been proposed. We have incorporated the above into the second paragraph of the Introduction.
Section1: The carbon emission targets in different fields are different, and their roles are also different. How does this affect the realization of the dual carbon targets?
Response:
Thank you for the helpful comments. We put this point in the introduction section. Indeed, the carbon emission targets in different fields are different, and their roles are also different. By comparing the carbon intensity of different industries, it is found that the carbon intensity of energy-intensive industries is much higher than that of other industrial sectors. The proportion of the total industrial output value of energy-intensive industries in the entire industrial sector has remained at nearly 30% in a long term, whereas the CO2 emissions accounted for approximately 80% of the entire industrial sector. Therefore, reducing the carbon intensity of energy-intensive industries in the future is one of the important contents to achieve the dual carbon goals.
Section1: Some data sources in the text are not very clear, please add relevant data source descriptions or references. (For example: carbon emissions or total GDP by region).
Response:
Thank you for raising this good point. This makes our paper more complete. We supplement the descriptions and sources of relevant data in the corresponding sections. According to your suggestion, we improved this paragraph as follows:
“3.5. Data source and description
The criteria for the definition of energy-intensive industries in “National Economic and Social Development Statistical Bulletin of China in 2019” are adopted in this study. The standard follows the national economic industry classification method, completely consistent with the industrial statistics published by the National Bureau of Statistics. The six major industries covered in the energy-intensive industry and their industry codes in the “National Economic Industry Classification (2011)” consist of petroleum refining and coking (25), chemical raw material and chemical product manufacturing (26), non-metallic mineral products (30), iron and steel (31), non-ferrous metals (32), as well as power industry (44). Data on carbon emissions of energy-intensive industries by province over the years can be downloaded freely from Carbon Emission Accounts and Data sets for emerging countries (CEADs) (https://www.ceads.net/data/province/). The total industrial output value of the energy-intensive industry originates from the “China Industrial Economic Statistical Yearbook” over the years, the consumption data of each energy by industry originate from the “China Energy Statistical Yearbook”, and the data of the remaining explanatory variables, such as GDP, population, FDI, R&D expenditure, number of enterprises, etc., come from the “China Statistical Yearbook”, “China Science and Technology Statistical Yearbook”, “China Environmental Yearbook”, “China Trade and Foreign Economic Statistical Yearbook” as well as statistical year-books of different provinces (https://tongji.oversea.cnki.net/oversea/engnavi/ navide-fault.aspx).Table 1 lists the basic statistical descriptions of the explanatory variables and the explained variables obtained. The time span of the basic data employed in this study is between 2000 and 2019, and the basic spatial unit is 30 provinces (Hong Kong, Macao, Taiwan and Tibet are not included due to missing data). Map vector data originate from National Geomatics Center of China (http://www.ngcc.cn/).”
Section2: How is carbon intensity defined? How do the development level, energy structure and industrial structure of a country affect carbon emission intensity?
Response:
Thank you for raising this question. How carbon intensity is defined is really important for this article. By consulting authoritative literature, carbon intensity can be expressed the resource utilization efficiency and carbon emission efficiency in economic development. The impact of a country's development level, energy structure and industrial structure on carbon emission intensity is very complex, and different scholars have drawn different conclusions on the degree and direction of the impact. Therefore, we provide a detailed review of relevant results in the literature review section. The modified content is as follows:
“The influencing mechanism behind carbon emission (intensity) spatiotemporal differences is complex and is caused by a variety of drivers. The factors studied by the existing research mainly include energy structure, industrial structure, economic output, population size, energy intensity, rate of urbanization, technological advancement, foreign trade, etc. Most studies show that economic growth will lead to increased carbon emissions [32-33]. On the one hand, the increase in residents’ income will prompt residents to live a high-energy-consuming lifestyle, resulting in increased carbon emissions; on the other hand, it will also prompt residents to realize the limitations of resources and environment, and make efforts to adopt energy-saving technologies to reduce carbon emission intensity [34]. Urbanization has maintained a positive correlation with carbon emissions for a long time, and is an essential factor in promoting China’s carbon emissions [35]. However, there is a certain debate about the direction of the effect of economic level and urbanization rate on carbon intensity. Some scholars believe that both of them have a negative correlation with carbon intensity, while some scholars believe that the two have a non-linear effect in addition to a direct linear effect on carbon intensity [36]. Technological progress is considered to be one of the most important factors in reducing carbon intensity. Typical domestic and foreign technologies include the reduction of fuel combustion in the process, processing conversion, and end use, as well as technologies such as carbon recovery, low-carbon energy, and carbon sinks [37-39]. Scholars generally believe that the industrial structure and energy structure contribute most to China’s carbon intensity effect, and that structural adjustment is still the main direction of China’s future emission reduction [40]. In addition, energy intensity is the main factor driving the reduction of China’s carbon intensity [41]. The impact of foreign direct investment on the host country's carbon intensity is twofold; carbon intensity can either be increased through the transfer of pollution-intensive industries or reduced by technology spillovers that make related production processes cleaner [42]. Guan et al. [43], Xu et al. [44], Ouyang and Lin [27] and other studies have shown that industrial activity is a key factor in increasing CO2 emissions. Xie et al. [45] found that energy-intensive industries such as power production, petroleum processing, coking, chemical products, metal smelting and rolling, and non-metallic mineral products have the most prominent impact on carbon emissions. For the above energy-intensive industries, Lin and Long [46] argue that energy intensity and energy mix may be beneficial for reducing CO2 emissions from China’s chemical industry. Yang et al. [30] found that growth in GDP and population per capita increases CO2 emissions from energy-intensive industries, while the share of service industries leads to a reduction in CO2 emissions.”
Section2: It is mentioned in the manuscript that "Technological progress is considered to be one of the most important factors in reducing carbon intensity, including both domestic and foreign technologies", but what kind of technology is not clearly explained in the manuscript.
Response:
We have made corresponding revisions to address your concerns. We have added a description of typical technologies for reducing carbon intensity in the corresponding section. Typical domestic and foreign technologies include the reduction of fuel combustion in the process, processing conversion, and end use, as well as technologies such as carbon recovery, low-carbon energy, and carbon sinks.
References:
- Wang, S.; Zeng, J.; Liu, X. Examining the multiple impacts of technological progress on CO2 emissions in China: a panel quantile regression approach. Renew. Sust. Energ. Rev. 2019, 103, 140-150.
- Li, P.; Ouyang, Y. Quantifying the role of technical progress towards China’s 2030 carbon intensity target. J. Environ. Plan. Manag. 2021, 64, 379–398.
- Zhang, L.W.; Sojobi, A.O.; Kodur, V.K.R.; Liew, K.M. Effective utilization and recycling of mixed recycled aggregates for a greener environment. J. Clean. Prod. 2019, 236, 117600.
Section2: Is the formula for calculating CO2 emissions for the 6 sub-sectors justified? What are the six sub-sectors? Also, how is the emission factor corrected? Please make it clearly.
Response:
In accordance with the “National Economic and Social Development Statistical Bulletin of China in 2019”, energy-intensive industries include 6 sub-sectors including non-metallic minerals, petroleum coking, electric power, chemicals, steel and non-ferrous metals. Regarding the formula for carbon dioxide emissions in the six sub-sectors, we refer to the existing research results. This is not made clear in the original text. Therefore, we added the latest references and data sources. We have made several revisions to address your comments:
“Data on carbon emissions of energy-intensive industries by province over the years can be downloaded freely from Carbon Emission Accounts and Data sets for emerging countries (CEADs) (https://www.ceads.net/data/province/).”
“We draw on similar expressions. Based on the existing research results on CO2 of energy-intensive industries [54-55], the EIICI is expressed as the amount of CO2 emitted per unit of industrial sales output value.”
References:
- Guan, Y.; Shan, Y.; Huang, Q.; Chen, H.; Wang, D.; Hubacek, K. Assessment to China’s Recent Emission Pattern Shifts. Earth’s Future 2021, 9, e2021EF002241.
- Liao, S.; Wang, D.; Xia, C.; Tang, J. China’s provincial process CO2 emissions from cement production during 1993–2019. Sci. Data 2022), 9(1), 1-14.
Section3: The derivation or source of some formulas in the manuscript is not very clear, please add relevant content
Response:
Thank you pointing this out. We checked all formulas in the manuscript, made adjustments where the derivation or source was not very clear, and added some references. First of all, the method for calculating carbon emissions of energy-intensive industries is deleted, which is based on the achievements of existing scholars, so there is no need to repeat it. Second, for the part of the spatial econometric model, we have added authoritative literature as support in the key places of formula derivation.
References:
- Wang, S.; Fang, C.; Wang, Y. Spatiotemporal variations of energy-related CO2 emissions in China and its influencing factors: An empirical analysis based on provincial panel data. Renew. Sust. Energ. Rev. 2016, 55, 505-515.
- Xu, L.; Chen, N.; Chen, Z. Will China make a difference in its carbon intensity reduction targets by 2020 and 2030?. Appl. Energy 2017, 203, 874-882.
Section4: The amount of economic purchases in the 20-year span is different. In addition, the extensive economic development method ten years ago and the current resource utilization are also different. Will this affect the conclusion?
Response:
Thank you for the helpful comments. Indeed, economic purchases do lead to discrepancies in regression results. By reviewing the literature, we found that similar studies have also discussed this issue, and it is believed that economic purchases have a limited impact on the overall general trend and conclusions. But this is indeed the defect of this paper, so we put this issue in the future study section as the content that needs to be further deepened. The relevant sections was revised accordingly as follows:
“New models and novel methods should be adopted to analyze the influencing mechanism of key drivers and the interaction between multiple drivers. This study primarily aims to reveal the influence direction, relative strength and spillover effect of different factors. Future research needs to further consider the bilateral causal relationship be-tween the dependent and independent variables to address the endogeneity problem. In the study of long-term series, it is necessary to further analyze the influence of factors such as economic purchases on the results. A geospatial weighted regression model should be used to explore the impact intensity of different factors in each province, in order to reveal the differences in the driving mechanisms between regions.”
Section5: How can the relevant calculations in the manuscript be validated in reality? Or how to justify the conclusion?
Response:
As the reviewer said, the relevant calculation results of the paper need to be roughly consistent with the actual development trend in order to verify the validity of the research results. In response to this problem, on the one hand, we compare with the results of existing similar studies, discuss the similarities and differences between the conclusions of this study and similar studies. On the other hand, it is discussed in future research that regional comparative studies can be conducted to examine whether relevant calculation results can be verified in typical case areas. The relevant sections was revised accordingly as follows:
“Some studies have found similar spatial spillover effects when studying the emissions of air pollutants and water pollutants [57,64]……Thus, companies are inclined to spend more money on efficient equipment [66]. A considerable number of energy-intensive enterprises are located in a single industrial park or industrial parks with relatively short distances, often forming energy-intensive industry agglomeration [67]……In response to this finding, it is not consistent from the research results of existing studies on the overall carbon intensity, which suggested that urbanization level significantly reduced carbon intensity [32]……”
“Finally, by conducting a regional comparative study, it is investigated whether the relevant calculation results can be verified in typical case areas.”

Reviewer 2 Report
This is a normative and meaningful study. A few comments for your reference:
(1) The marginal contribution of research needs to be further clarified. As a matter of fact, there have been many studies on double carbon using medium macro data, and the marginal contribution of this study is not clear. It does not mean that few people have paid attention to this topic, which does not constitute a new marginal contribution.
(2) The study needs to be further compared with similar studies. That is to add a discussion section to discuss the similarities and differences between this study and similar studies and the reasons for the differences.
(3) There may be endogeneity problems in the study, which should be discussed by the author.
Author Response
Dear Reviewer:
Thank you for your comments concerning our manuscript. Those comments are all valuable and very helpful for revising and improving our paper, as well as the important guiding significance to our research. We have revised our paper carefully to address the comments and have made many corrections in the original text in revision mode, which we hope have adequately addressed your concerns. We look forward to hearing from you.
Best regards,
Authors
Comments and Responses:
This is a normative and meaningful study. A few comments for your reference:
(1) The marginal contribution of research needs to be further clarified. As a matter of fact, there have been many studies on double carbon using medium macro data, and the marginal contribution of this study is not clear. It does not mean that few people have paid attention to this topic, which does not constitute a new marginal contribution.
Response:
Thank you for the valuable suggestion. Through the background analysis and literature review in the introduction, we clarify the marginal contribution of this paper from two aspects: the lack of discussion of the spatial differences in carbon intensity of energy-intensive industries and the lack of attention to its spillover effects. The relevant sections was revised accordingly as follows:
“The extensive studies have provided a very useful reference for the text, whereas the research on energy-intensive industries still has deficiencies. First, existing studies lack the exploration of the spatial differences in carbon intensity of energy-intensive industries, so it is difficult to provide help for the formulation of scientific regional decomposition plans in the process of emission reduction in this key industry. Second, spatial elements often have spillover effects and the first law of geography also highlights that the closer the spatial distance between things, the stronger the interdependence will be [53]. Most of the existing research has recognized regions as separate individuals, and the spatial dependence and spatial heterogeneity of carbon intensity have been rarely discussed from the perspective of geography and spatial interaction. In this way, it is difficult to reveal the spatial correlation and provide a scientific basis for proposing emission reduction measures for joint prevention and control. To fill this research gap, this study first finds the characteristics of spatiotemporal differences in EIICI, and then uses the spatial econometric method based on the STIRPAT model to investigate the drivers and spatial spillover effects of EIICI.”
(2) The study needs to be further compared with similar studies. That is to add a discussion section to discuss the similarities and differences between this study and similar studies and the reasons for the differences.
Response:
We really appreciate your helpful suggestion. We performed a comparison of similar studies for some important results. According to your suggestion, the relevant sections was revised as follows:
“Some studies have found similar spatial spillover effects when studying the emissions of air pollutants and water pollutants [57,64]……Thus, companies are inclined to spend more money on efficient equipment [66]. A considerable number of energy-intensive enterprises are located in a single industrial park or industrial parks with relatively short distances, often forming energy-intensive industry agglomeration [67]……After the local environmental control is optimized, energy-intensive enterprises face increased pollution control costs. Instead of investing in advanced technology and environmental protection equipment to increase cleaning efficiency, some companies have been relocated to neighboring provinces with looser environmental regulations [57]…..In response to this finding, it is not consistent from the research results of existing studies on the overall carbon intensity, which suggested that urbanization level significantly reduced carbon intensity [32]……”
(3) There may be endogeneity problems in the study, which should be discussed by the author.
Response:
Thank you for raising this good point. The endogenous problem that may stem from the bilateral causal relationship between the dependent and independent variables as well as the neglected variables might lead to biased estimations. However, this paper does not discuss this issue in depth, and only assumes a one-way relationship between the dependent variable and the independent variable. Admittedly, this is the shortcoming of this article. We have left this issue in the discussion for further exploration in the future. We have also made corresponding revisions in the paper:
“New models and novel methods should be adopted to analyze the influencing mechanism of key drivers and the interaction between multiple drivers. This study primarily aims to reveal the influence direction, relative strength and spillover effect of different factors. Future research needs to further consider the bilateral causal relationship between the dependent and independent variables to address the endogeneity problem. In the study of long-term series, it is necessary to further analyze the influence of factors such as economic purchases on the results. A geospatial weighted regression model should be used to explore the impact intensity of different factors in each province, in order to reveal the differences in the driving mechanisms between regions.”

Reviewer 3 Report
The objective of manuscript entitled “What cause large spatiotemporal differences in carbon intensity of energy-intensive industries in China? Evidence from provincial data during 2000-2019” is to study the spatiotemporal differences and drivers of carbon intensity in different regions of China. The paper is well written. However, the authors make careless mistake on collecting the raw data and apply into the model, leading to the data interpretation is wrong based on the wrong calculated data. In line 222, the primary data used by the authors came from only ONE author “Shun et al., (50)”. The data collected about carbon dioxide emission is from 1997-2015 only. How’s the author reflected the situation in 2019 or even this year? Besides, I believe that “Shun et al” is only the only source of data. For example:
Q Ma, M Tariq, H Mahmood, Z Khan - Technology in Society, 2022. The nexus between digital economy and carbon dioxide emissions in China: The moderating role of investments in research and development
KH Wang, JM Kan, CF Jiang, CW Su - Sustainability, 2022. Is Geopolitical Risk Powerful Enough to Affect Carbon Dioxide Emissions? Evidence from China
X Yang, D Wang - Sustainability, 2022. Heterogeneous Environmental Regulation, Foreign Direct Investment, and Regional Carbon Dioxide Emissions: Evidence from China
For the paper published in the year of 2022, the author should use updated information instead of using the data collected in the year of 2015!
The author should collect more data from other authors and make prediction and calculation again. Therefore, I think author should recalculate ALL data and interpret the results so as to produce meaningful data and update information to all readers.
Below is the specific comments to the manuscript:
L52: Elaborate the phrase “in the remaining time”
L87-89: The statement is weak. For scientific paper, we should focus on scientific reasons rather than the monetary reasons. Besides, I found authors tried to apply model based on the GDP of the region in China (i.e. Table 1) however the author pointed out the disadvantage of using GDP to calculate carbon intensity. The point is controversial. Please explain.
L104: What is “rich” means?
L118-119: Other than east and west, how about the status of development in southern and northern part of China?
L153: The foreign investment policy has been implemented in China since 1980s, therefore I don’t think “energy intensity and foreign investment is the main factors driving the reduction of carbon intensity.”. Do you know the concepts of NIMBY? Can this concept applied into these situation in China so that China government should amend the policies about carbon dioxide reduction?
L164-190: I think it is not necessary to explain other methods of modelling. Please delete the paragraph. Instead, please review critically the measurement of “EIICI” used to calculate carbon intensity in China or other countries because the measurement of “EIICI” is the main theme of these manuscript.
L202: What is “the first law of Geography”? Any citation? Is the law created by your research team?
For the rest of the part in the manuscript, I think I cannot give any comments until the authors treat the data and make calculation and prediction again.
Author Response
Dear Reviewer:
Thank you for your comments concerning our manuscript. Those comments are all valuable and very helpful for revising and improving our paper, as well as the important guiding significance to our research. We have revised our paper carefully to address the comments and have made many corrections in the original text in revision mode, which we hope have adequately addressed your concerns. We look forward to hearing from you.
Best regards,
Authors
Comments and Responses:
The objective of manuscript entitled “What cause large spatiotemporal differences in carbon intensity of energy-intensive industries in China? Evidence from provincial data during 2000-2019” is to study the spatiotemporal differences and drivers of carbon intensity in different regions of China. The paper is well written.
However, the authors make careless mistake on collecting the raw data and apply into the model, leading to the data interpretation is wrong based on the wrong calculated data. In line 222, the primary data used by the authors came from only ONE author “Shun et al., (50)”. The data collected about carbon dioxide emission is from 1997-2015 only. How’s the author reflected the situation in 2019 or even this year? Besides, I believe that “Shun et al” is only the only source of data.
For the paper published in the year of 2022, the author should use updated information instead of using the data collected in the year of 2015! The author should collect more data from other authors and make prediction and calculation again. Therefore, I think author should recalculate ALL data and interpret the results so as to produce meaningful data and update information to all readers.
Response:
Sorry about the confusing expression. There is no problem with data sources for carbon emissions from energy-intensive industries. However, we did not explain clearly. The carbon emissions data in this paper are derived from Carbon Emission Accounts and Data sets for emerging countries (CEADs). The database contains carbon emission data by province and industry in China from 1997 to 2019. The original manuscript cited Author "Shan et al., (50)" to illustrate the calculation of carbon emissions in that database. But as the reviewer pointed out, the data in that document only refers to 1997-2015. Therefore, we made 3 modifications to this problem. The first is to delete the description of the calculation method of carbon emissions in CEADs in the Methods section. Second, the key information about carbon emissions in CEADs is described in the data source section. The third is to delete the document "Shan et al., (50)", and add two newest documents that directly describe the database and calculation methods. The relevant sections were revised accordingly as follows:
In the “Measurement of EIICI” section: “Based on the existing research results on CO2 of energy-intensive industries [54-55], the EIICI is expressed as the amount of CO2 emitted per unit of industrial sales output value.”
In the “Data source and description” section: “Data on carbon emissions of energy-intensive industries by province over the years can be downloaded freely from Carbon Emission Accounts and Data sets for emerging countries (CEADs) (https://www.ceads.net/data/province/).”
References:
- Guan, Y.; Shan, Y.; Huang, Q.; Chen, H.; Wang, D.; Hubacek, K. Assessment to China’s Recent Emission Pattern Shifts. Earth’s Future 2021, 9, e2021EF002241.
- Liao, S.; Wang, D.; Xia, C.; Tang, J. China’s provincial process CO2 emissions from cement production during 1993–2019. Sci. Data 2022), 9(1), 1-14.
Below is the specific comments to the manuscript:
L52: Elaborate the phrase “in the remaining time”
Response:
Thank you pointing this out. What we want to express here is that between now and 2030, there is limited room for reducing carbon intensity through industrial restructuring. Indeed, the expression "in the remaining time" is not clear. The modified sentence is as follows:
“Since China has vigorously adjusted its industrial structure over the past decade, which has led to significantly reduced carbon intensity, there has been limited room for reducing carbon intensity through industrial restructuring towards 2030 [8].”
L87-89: The statement is weak. For scientific paper, we should focus on scientific reasons rather than the monetary reasons. Besides, I found authors tried to apply model based on the GDP of the region in China (i.e. Table 1) however the author pointed out the disadvantage of using GDP to calculate carbon intensity. The point is controversial. Please explain.
Response:
Thank you for the helpful advice. This statement is indeed not very relevant to our research topic. Therefore, we have deleted this statement from the original text. In addition, it is inappropriate to use GDP to calculate the carbon intensity of energy-intensive industries. Therefore, based on the existing research results on CO2 of energy-intensive industries, the EIICI is expressed as the amount of CO2 emitted per unit of industrial sales output value. Please refer to the "Calculation of EIICI" section for this modification.
L104: What is “rich” means?
Response:
Sorry about the confusing expression. This sentence has been revised as follows: ”… …and these studies have achieved fruitful results.”
L118-119: Other than east and west, how about the status of development in southern and northern part of China?
Response:
Regarding the spatial differences in carbon emissions or carbon intensity in China, most scholars focus on the differences between the eastern, central, and western regions, and seldom consider the differences between southern and northern China. This paper takes note of this, taking the difference between the south and the north as an important feature. The relevant content is as follows:
“Furthermore, from the perspective of the differences between the north and the south, ten northern provinces of Xinjiang, Gansu, Inner Mongolia, Ningxia, Shaanxi, Shanxi, Hebei, Liaoning, Jilin and Heilongjiang are taken in this study as a whole to examine the difference in EIICI between the northern provinces and the rest of the southern provinces. In 2000, the average EIICI of northern provinces and other provinces was 15.73 t/104 yuan and 7.70 t/104 yuan, respectively, the former was 2.04 times that of the latter. In 2019, the average EIICI of the two was regulated to 4.24 t/104 yuan and 1.66 t/104 yuan, respectively, and the gap between the former and the latter increased to 2.55 times.”
L153: The foreign investment policy has been implemented in China since 1980s, therefore I don’t think “energy intensity and foreign investment is the main factors driving the reduction of carbon intensity”. Do you know the concepts of NIMBY? Can this concept applied into these situation in China so that China government should amend the policies about carbon dioxide reduction?
Response:
Thank you for the helpful comments. Yeah, foreign direct investment does have a " Pollution Haven " effect. Developed countries or regions transfer high-carbon industries to other countries or regions (usually developing countries, such as China) through foreign investment, which will lead to more serious pollution in developing countries. This effect is caused by the transfer of pollution-intensive industries, and multinational companies in developed countries choose to locate their pollution-intensive industries or production links in China, which increases the carbon emissions of the host country. Of course, on the other hand, there is also a "pollution halo" effect. FDI brings advanced technology to the host country, making the related production process cleaner and helping to reduce carbon intensity. This effect is generated by technology spillovers, with multinational corporations bringing better technology and positive spillovers to domestic firms. Technology spillovers from FDI contribute to emissions reductions in FDI destinations and surrounding areas. Therefore, consider the duality of the impact of FDI on the host's carbon emissions. The Chinese government's policy should consider environmental capacity when attracting foreign investment, and it is best to direct FDI to areas with high environmental capacity for deployment. Thank you for correcting. After reviewing some existing studies, we modified this sentence as follows:
“The impact of foreign direct investment on the host country's carbon intensity is twofold; carbon intensity can either be increased through the transfer of pollution-intensive industries or reduced by technology spillovers that make related production processes cleaner [42].”
References:
- Lin, H.; Wang, X.; Bao, G.; Xiao, H. Heterogeneous Spatial Effects of FDI on CO2 Emissions in China. Earth’s Future 2022, 10, e2021EF002331.
L164-190: I think it is not necessary to explain other methods of modelling. Please delete the paragraph. Instead, please review critically the measurement of “EIICI” used to calculate carbon intensity in China or other countries because the measurement of “EIICI” is the main theme of these manuscript.
Response:
We have made correction according to the Reviewer’s comments. First, we no longer describe "other methods of modelling" in detail and delete relevant content. Second, with regard to the calculation method of carbon intensity, most of the indicators used in existing studies are the ratio of carbon emissions to economic output, which has a wide range of applicability. We draw on similar expressions. The relevant sections was revised accordingly as follows:
“Regarding the selection of indicators of carbon intensity, most of the existing studies use the ratio of carbon emissions to economic output, that is, the carbon dioxide emitted per unit of economic output [5,24,26,52]. We draw on similar expressions. Based on the existing research results on CO2 of energy-intensive industries, the EIICI is expressed as the amount of CO2 emitted per unit of industrial sales output value.”
References:
- Han, Y.; Jin, B.; Qi, X.; Zhou, H. Influential Factors and Spatiotemporal Characteristics of Carbon Intensity on Industrial Sectors in China. Int. J. Environ. Res. Public Health 2021, 18, 2914.
- Wang, Z.; Zhang, B.; Liu, T. Empirical analysis on the factors influencing national and regional carbon intensity in China. Renew. Sust. Energ. Rev. 2016, 55, 34-42.
- Cheng, Y.; Wang, Z.; Ye, X.; Wei, Y. D. Spatiotemporal dynamics of carbon intensity from energy consumption in China. J. Geogr. Sci. 2014, 24(4), 631-650.
- Xu, L.; Chen, N.; Chen, Z. Will China make a difference in its carbon intensity reduction targets by 2020 and 2030?. Appl. Energy 2017, 203, 874-882.
L202: What is “the first law of Geography”? Any citation? Is the law created by your research team?
Response:
Thank you pointing this out. “The first law of Geography” means that everything is related to everything else, but near things are more related to each other. We have added an authoritative reference.
References:
- Tobler, W. On the first law of geography: A reply. Ann. Assoc. Am. Geogr. 2004, 94, 304–310.

Reviewer 4 Report
1. Kindly state the aim and objectives of your study in the introduction section.
2. Please include a flow chart to describe your study.
“The carbon intensity of energy-intensive industries (EIICI) is significantly higher than that of other industrial sectors, which is recognized as the top priority of China’s future emission reduction actions”.
3. In tabular format, list provinces classified under eastern region, central region and western region.
4. Do not remove Figure 2. To further understand the figure, also plot the EIICI using bar charts for easy presentation and understanding.
5. Investigate variables which have the most significant effect on EIICI. Rank the variables in the form of EIICI using bar charts or each province. Discuss the reasons for your results.
6. State and discuss the health and environmental implications of EIICI based on literature.
7. Based on your results in 5 and 6, discuss relevant policy measures to reduce EIICI.
8. Kindly improve your introduction by citing these papers as they relate to climate change and carbon emission:
L.W. Zhang, A.O. Sojobi, V.K.R. Kodur, K.M. Liew. Effective utilization and recycling of mixed recycled aggregates for a greener environment. Journal of Cleaner Production, 236 (2019) 1-27
Adegoke C.W., Sojobi AO. Climate change impact on infrastructure in Osogbo metropolis, south-west Nigeria. Journal of Emerging Trends in Engineering and Applied Sciences, 6, 3, (2015) 156-165
Author Response
Dear Reviewer:
Thank you for your comments concerning our manuscript. Those comments are all valuable and very helpful for revising and improving our paper, as well as the important guiding significance to our research. We have revised our paper carefully to address the comments and have made many corrections in the original text in revision mode, which we hope have adequately addressed your concerns. We look forward to hearing from you.
Best regards,
Authors
Comments and Responses:
(1) Kindly state the aim and objectives of your study in the introduction section.
Response:
We really appreciate your helpful suggestion. According to your suggestion, we improved the paragraph as follows:
“Accordingly, the conclusions obtained only from the research at the overall level of the country are difficult to apply to the development of different regions, which has become a significant issue that has attracted wide attention from the government and scholars. Therefore, focusing on reducing the EIICI, using the panel data of 30 provinces in China from 2000 to 2019, this paper characterizes the temporal and spatial differences of EIICI. On this basis, further considering the spatial spillover effect, several spatial econometric models are established to reveal the driving factors causing the differences. Hopefully, the research results can provide a reference for government departments to formulate and implement differentiated and targeted regional policies.”
(2) Please include a flow chart to describe your study.
“The carbon intensity of energy-intensive industries (EIICI) is significantly higher than that of other industrial sectors, which is recognized as the top priority of China’s future emission reduction actions”.
Response:
Thank you pointing this out. However, the above statement comes from the conclusions of existing research. Due to the lack of statistics for other industries, it is difficult for us to provide a comparison chart for each industry. We therefore supplement the following two references to support the above statement.
References:
- Liu, H.C.; Fan, J.; Zhou, K.; Wang, Q. Exploring regional differences in the impact of high energy-intensive industries on CO2 emissions: Evidence from a panel analysis in China. Environ. Sci. Pollut. Res. 2019, 26(25), 26229-26241.
- Zhang, X.; Lin, M.; Wang, Z.; Jin, F. The impact of energy-intensive industries on air quality in China’s industrial agglomerations. J. Geogr. Sci. 2021, 31(4), 584-602.
(3) In tabular format, list provinces classified under eastern region, central region and western region.
Response:
Considering the Reviewer’s suggestion, we have list provinces classified under eastern region, central region and western region, as shown in Table 2.
Table 2. Distribution of the three major regions in China
Regions |
Provinces |
Eastern region |
Beijing, Tianjin, Hebei, Liaoning, Shanghai, Jiangsu, Zhejiang, Fujian, Shandong, Guangdong, Hainan |
Central region |
Shanxi, Jilin, Heilongjiang, Anhui, Jiangxi, Henan, Hubei, Hunan |
Western region |
Inner Mongolia, Guangxi, Chongqing, Sichuan, Guizhou, Yunnan, Shannxi, Gansu, Qinghai, Ningxia, Xinjiang |
(4) Do not remove Figure 2. To further understand the figure, also plot the EIICI using bar charts for easy presentation and understanding.
Response:
Considering the Reviewer’s suggestion, after comparing the bar chart and the line chart, we try to use the line chart to draw the EIICI of each province from 2000 to 2019 for easy presentation and understanding. As shown in the figure below.
(5) Investigate variables which have the most significant effect on EIICI. Rank the variables in the form of EIICI using bar charts or each province. Discuss the reasons for your results.
Response:
Focusing on the research objectives of this paper, the spatial econometric model we adopted mainly addresses the spatial spillovers and direct and indirect effects of drivers. Unfortunately, bar charts cannot be used to rank the impact intensity of different variables in each province. Thank you for your positive and constructive comments. In future research, we can use the geospatial weighted regression model to achieve the above goals. We have written this into the future study Section.
(6) State and discuss the health and environmental implications of EIICI based on literature.
Response:
The carbon intensity of energy-intensive industries is much higher than that of other industrial sectors, which is the top priority of China's future emission reduction actions. Therefore, if EIICI is maintained at a high level for a long time, the greenhouse gases it produces will directly endanger people's health and quality of life. We have incorporated this content into the Introduction Section as an important background to underpin the significance of our study.
References:
- Zhang, X.; Lin, M.; Wang, Z.; Jin, F. The impact of energy-intensive industries on air quality in China’s industrial agglomerations. J. Geogr. Sci. 2021, 31(4), 584-602.
- Schönsleben, P.; Vodicka, M.; Bunse, K.; Ernst, F. O. The changing concept of sustainability and economic opportunities for energy-intensive industries. CIRP annals. 2010, 59(1), 477-480.
(7) Based on your results in 5 and 6, discuss relevant policy measures to reduce EIICI.
Response:
We have made corresponding revisions to address your concerns. In the last section, we have added a sub-heading of Policy Implications to discuss relevant policy measures to reduce EIICI. It is necessary to emphasize that policymakers should design differentiated abatement policies based on dominant factors, spatial effects, regional differences instead of applying similar policies to all provinces. The relevant content is as follows:
“6.2. Policy implications
The above results are of high implications for policy. First, the dominant drivers affecting the EIICI should be determined, as well as the key direction of emission re-duction policies. Emphasize the role of technological innovation in reducing EIICI, and combine independent innovation with the introduction of advanced foreign industrial technology and management experience. In order to adjust the energy consumption structure of high-energy-consuming industries and increase the share of clean energy consumption, the government needs to provide more incentives (such as increasing environmental taxes and increasing clean energy subsidies). Given the characteristics of most high-energy-consuming enterprises with small scale and scattered locations, we should fully exploit the scale effect, optimize the industrial space layout and facilitate the agglomerated development of energy-intensive industries. It is necessary to strengthen environmental control, formulate strict emission standards, and appropriately increase punitive damages for companies exceeding the standards. Second, targeted emission reduction measures should be formulated based on the reality of regional differences. In the eastern coastal areas, the independent innovation and the introduction of foreign advanced technologies should be facilitated, the application of innovative technologies in industrial production should be boosted, and ecological industrial clusters and a circular economy should be created. The central and western regions and northern provinces are required to largely optimize the energy consumption structure and progressively reduce the high consumption and high emission development mode dominated by coal. In the meantime, the transfer of energy-intensive industries from the eastern region should be actively and effectively dealt with to avoid becoming a “pollution heaven”. Local governments in the central and western regions should strengthen environmental supervision and rationally facilitate industrial layout under the premise that the carrying capacity of resources and environment is evaluated. Third, stress should be placed on the spatial spillover effect of EIICI, and regional coordinated development and control strategies should be formulated. In particular, when formulating economic development policies, the governments should consider the transfer of carbon emissions to emphasize the inhibition of local EIICI, as well as to consider the effect on surrounding areas. To reduce the overall carbon intensity of the region, administrative boundaries should be broken, and a cooperative governance model of overall planning, resource sharing, industrial collaboration and information sharing should be implemented in adjacent regions.”
(8) Kindly improve your introduction by citing these papers as they relate to climate change and carbon emission:
Response:
Thank you for providing these two valuable references. We have cited these two references where appropriate.
References:
- Zhang, L.W.; Sojobi, A.O.; Kodur, V.K.R.; Liew, K.M. Effective utilization and recycling of mixed recycled aggregates for a greener environment. J. Clean. Prod. 2019, 236, 117600.
Adegoke, C.W.; Sojobi, A.O. Climate change impact on infrastructure in Osogbo metropolis, south-west Nigeria. J. Emerg. Trends Eng. Appl. Sci. 2015, 6(3), 156-165.

Round 2
Reviewer 2 Report
I have no other comments, thank you.
Reviewer 3 Report
The revised manuscript showed substantial improved including the content and citations. I haven't had any further comments.
Reviewer 4 Report
Well done for the great revision!
This manuscript is a resubmission of an earlier submission. The following is a list of the peer review reports and author responses from that submission.